# Supplementation of Schisandrin B in Semen Extender Improves Quality and Oxidation Resistance of Boar Spermatozoa Stored at 4 °C

**DOI:** 10.3390/ani13050848

**Published:** 2023-02-25

**Authors:** Yunfa Xie, Zhiying Chen, Yanling Wang, Xiayun Peng, Ni Feng, Xiaoye Wang, Yinsheng Tang, Xun Li, Chunrong Xu, Chuanhuo Hu

**Affiliations:** 1College of Animal Science and Technology, Guangxi University, Nanning 530004, China; 2Guangxi Key Laboratory of Animal Reproduction, Breeding and Disease Control, Nanning 530004, China; 3Animal Husbandry Research Institute, Guangxi Vocational University of Agriculture, Nanning 530007, China; 4Guangxi Work Station of Livestock & Poultry Breed Improvement, Nanning 530001, China

**Keywords:** schisandrin B, sperm quality, boar, oxidative stress

## Abstract

**Simple Summary:**

Sch B is the active component extracted from Sch B Fructus. It shows reproductive protection and spermatogenesis effects in animals. As far as we know, no research has been conducted with Sch B on boar semen preservation at 4 °C. Therefore, the current research aimed to explore the effects of Sch B on quality, oxidation resistance, and decapacitation of boar spermatozoa at 4 °C. The results showed that the positive effects of 10 μmol/L Sch B were obvious for boar semen preservation at 4 °C.

**Abstract:**

During cold storage, boar spermatozoa undergo oxidative stress, which can impair sperm function and fertilizing capacity. The objective of the present study was to assess the effects of Schisandrin B (Sch B) in semen extenders on the quality of boar semen stored at hypothermia. Semen was collected from twelve Duroc boars and diluted in extenders supplemented with different concentrations of Sch B (0 μmol/L, 2.5 μmol/L, 5 μmol/L, 10 μmol/L, 20 μmol/L, and 40 μmol/L). Here, we demonstrated that 10 μmol/L Sch B provided the best effects on motility, plasma membrane integrity, acrosome integrity, sperm normality rate, average movement velocity, wobbility, mitochondrial membrane potential (MMP), and DNA integrity of sperm. The results of Sch B effects on antioxidant factors in boar sperm showed that Sch B significantly elevated the total antioxidant capacity (T-AOC) and markedly decreased the reactive oxygen species (ROS) and malondialdehyde (MDA) content of sperm. The expression of catalase (CAT) and superoxide dismutase (SOD) mRNA was increased, while the expression of glutathione peroxidase (GPx) mRNA demonstrated no change compared to non-treated boar sperm. Compared to the non-treated group, Sch B triggered a decrease in Ca^2+^/protein kinase A (PKA) and lactic acid content in boar sperm. Similarly, Sch B led to a statistically higher quantitative expression of AWN mRNA and a lower quantitative expression of porcine seminal protein I (PSP-I) and porcine seminal protein II (PSP-II) mRNA. In a further reverse validation test, no significant difference was observed in any of the parameters, including adhesion protein mRNA, calcium content, lactic acid content, PKA, and protein kinase G (PKG) activity after sperm capacitation. In conclusion, the current study indicates the efficient use of Sch B with a 10 μmol/L concentration in the treatment of boar sperm through its anti-apoptosis, antioxidative, and decapacitative mechanisms, suggesting that Sch B is a novel candidate for improving antioxidation and decapacitation factors in sperm in liquid at 4 °C.

## 1. Introduction

At present, the preservation of pig semen mainly includes chill storage (4 °C) and ultra-cryopreservation (−196 °C). Cryopreservation results in the low fertility of pig semen, which has a serious impact on research in the boar breeding industry. The pregnancy rate and litter size are similar at 4 °C fluid storage compared with natural mating. Therefore, liquid storage at 4 °C is an effective method to preserve boar semen. In mammals, boar sperm is susceptible to temperature due to its special membrane structure. Especially under low-temperature conditions, the sperm membrane structure is easily destroyed, resulting in ultrastructural changes in the acrosome and plasma membrane, including deterioration of sperm motility, viability, and DNA integrity, resulting in reduced sperm quality and thus reduced sperm longevity and fertility. Even if appropriate chilling and cryopreservation protocols have already been developed for many species, as shown in bull [1,2], ram [3,4], goat [5], and mouse [6] studies. There are limited data on the successful freezing, liquid storage, and cryopreservation of boar sperm. The cryopreservation of boar semen is still in the research and improvement stage; it is more commonly used in the preservation of sperm of threatened species and has not been widely used, which makes the cryopreservation research of boar semen of very important practical significance. Oxidative stress is an important factor affecting sperm quality during the liquid storage of boar semen in vitro. Adding various protective components to the diluent is an effective method to maintain the survival and function of sperm.

Schisandrin B (Sch B) is mainly derived from Schisandra chinensis (Turcz) Baillon and has the highest content in Schisandra chinensis. Sch B, C_23_H_28_O_6_ with a molecular weight of 400.6, is a natural antioxidant that not only enhances the antioxidant status of cells but also has been found to prevent the structural and functional degradation of cell mitochondria, both of which are critical to cell survival. Experimental investigations have demonstrated that Sch B has multiple pharmacological properties, including anti-oxidation, anti-inflammation [7], anti-apoptosis [8], and anti-tumor [9] effects. Meanwhile, research suggests that Sch B may be a potential therapeutic agent against anxiety associated with oxidative stress through reduced malondialdehyde (MDA) levels, production of reactive oxygen species (ROS), and neuronal damage [10]. In terms of reproductive function, Sch B has spermatogenic effects and can repair damaged seminiferous tubules and spermatogenic cells in the testis, providing a novel way to treat male infertility [11]. Based on its antioxidant and reproductive function protection properties, we speculate that Sch B could be used for sperm protection during liquid preservation of boar semen at 4 °C.

To date, the antioxidative effects of Sch B in the preservation of boar sperm and the mechanism of protective effects by Sch B have not been determined. In this context, Sch B is used for the first time in this study to explore its effects on boar sperm liquid preservation, revealing the underlying mechanism(s) of Sch B in sperm quality parameters, antioxidant capacity, and sperm fertilization ability during liquid preservation of boar semen at 4 °C. We further evaluated the effects of Sch B on sperm capacitation parameters before and after capacitation.

## 2. Materials and Methods

### 2.1. Chemicals

Unless otherwise stated, all chemicals used in this study were purchased from Solarbi (Beijing, China). Sch B was obtained from Victory Biological Technology (Cat. No. 61281-37-6, Sichuan Province, China).

### 2.2. Collection and Processing of the Ejaculates

Mixed semen from 12 mature Duroc boars aged 2–4 years was collected through the hand-gloved technique. Boars were fed at the Keda Livestock and Poultry Improvement Co., Ltd. (Guangxi Province, China). An entire ejaculation was collected twice a week. Only ejaculates that satisfied the sperm quality requirements to produce commercial semen AI-doses (namely, sperm concentration >200 × 10^6^ sperm/mL, sperm motility >70%, and morphologically normal sperm >75%) were included in the study. For seminal plasma harvesting, entire ejaculates were centrifuged twice (1500× *g* for 10 min at room temperature (Rotofix 32A, Hettich Centrifuge UK, Newport Pagnell, Buckinghamshire, UK)).

### 2.3. Experimental Designs

In this study, three experiments were carried out.

#### 2.3.1. Experiment I: Effects of Different Concentrations of Sch B on Sperm Quality Parameters

Boar semen samples were diluted in the extender (glucose 3 g, sucrose 4 g, and egg yolk 20 mL in double-distilled water and volume made up to 100 mL) to a concentration of 80 million sperm/mL. Sch B was added to yield six different final concentrations: 0, 2.5, 5, 10, 20, and 40 μmol/L. The extended sperm were incubated at 4 °C for 24 h. The sperm motility, abnormal sperm morphology, average movement velocity, wobbility (wobbility is the measure by which the sperm head wobbles along the average spatial path of its actual trajectory), acrosome integrity, plasma membrane integrity, mitochondrial membrane potential (MMP), and DNA integrity were assessed. Each experiment was repeated three times.

#### 2.3.2. Experiment II: Effects of Sch B on Antioxidant Capacity of Boar Sperm after Low Temperature Preservation

In experiment 2, we measured sperm superoxide dismutase (SOD), catalase (CAT) and glutathione peroxidase (GPx) mRNA expression, total antioxidant capacity (T-AOC) ability, ROS content, MDA content, Ca^2+^ content, lactic acid content, the mRNA expression of AWM, AQN1, AQN3, porcine seminal protein I (PSP-I), porcine seminal protein II (PSP-II), protein kinase A(PKA), and protein kinase G (PKG) activity before capacitation. Each experiment was repeated three times.

#### 2.3.3. Experiment III: Effects of 10 μmol/L Sch B Supplementation to Boar Semen Dilution on Sperm Quality after Sperm Capacitation

We measured Ca^2+^ content, lactic acid content, the mRNA expression of AWM, AQN1, AQN3, PSP-I, PSP-II, PKA, and PKG activity after sperm capacitation. Each experiment was repeated three times.

### 2.4. Methods

#### 2.4.1. Motility, Abnormal Sperm Morphology, Average Movement Velocity and Wobbility Analyzed with Computer-Assisted Semen Analysis (CASA)

After rewarming, 1 mL sperm was centrifuged at a speed of 900× *g*/min in a cryogenic centrifuge at 4 °C for 5 min, washed with PBS, repeated 3 times, and then suspended with PBS to adjust the sperm density to 1 × 10^6^~2 × 10^6^ sperm/mL. Then, 20 μL of the sperm resuspended in PBS was applied to a slide, covered with a coverslip, and the sperm motility, abnormal sperm morphology, average movement velocity, and wobbility were examined by CASA (CEROSII, IMV, Shanghai, China) after low-temperature preservation.

#### 2.4.2. Plasma Membrane Integrity of Spermatozoa by HOST

A sample of 1 mL of rewarmed sperm was taken and centrifuged with phosphate-buffered saline (PBS) at 900× *g*/min in a low-temperature centrifuge at 4 °C for 5 min. This was repeated three times and added to the pre-configured hypotonic solution. The sperm density was adjusted to 1 × 10^6^~2 × 10^6^ sperm /mL, and incubated at 37 °C for 30 min. Then, 15 μL of the mixture was put on a slide, covered with a cover slide, and the tail bending rate of sperm was observed under a microscope at 400 times. At least 200 sperm were counted, and this was repeated 3 times.

#### 2.4.3. Acrosome Integrity of Sperm by Coomassie Brilliant Blue

After sperm rewarming, the sperm density was adjusted to 1 × 10^6^~2 × 10^6^ sperm /mL, and the semen was washed by PBS centrifugation 3 times. The sperm were fixed with formalin and sodium citrate in the EP tube. The fixed sperm 2000× *g*/min was centrifuged for 5 min and then the supernatant was discarded. The sperm were washed once with PBS. Then, 100 μL of PBS was added to the sperm to make a suspension. A 10 μL mixed smear was taken and dried naturally. The air-dried slides were stained in a staining jar with Coomassie brilliant blue, and then the staining solution was slowly rinsed off with water and placed on a staining rack to dry naturally. The acrosome integrity was observed under the microscope (acrosomes in intact sperm were stained with Coomassie brilliant blue to a smooth and uniform blue surface). The experiment was repeated 3 times.

#### 2.4.4. Detection of Sperm DNA Damage Rate

The operation was carried out under dark conditions according to the instructions of the AO fluorescent staining kit (Cat. DA0037, Leagene, Beijing, China). After staining, the same samples were examined under a fluorescence microscope (the stained yellow-green fluorescence was the nucleus of normal sperm, and the green granules were the nucleus of DNA damage). At least 200 spermatozoa were counted per smear. The experiment was repeated 3 times.

#### 2.4.5. MMP in Spermatozoa

The sperm density was adjusted to (1–2) × 10^6^ sperm /mL. The sperm were rewashed with PBS by centrifugation (900× *g*; 5 min), and the supernatant was discarded. The operation was performed according to the MMP detection kit (JC-1) (Cat. CA1310, Solarbio, Beijing, China), and the fluorescence values of each group were detected by a multi-functional microplate tester. The experiment was repeated 3 times.

#### 2.4.6. Measurement of Oxidant-Antioxidant Status in Boar Sperm

After being rewarmed, the sperm density was adjusted to (1–2) × 10^6^ sperm /mL. The semen was centrifuged 3 times with PBS. Oxidant–antioxidant parameters (MDA (Cat. BC0025, Solarbio, Beijing, China), ROS (Cat. S0033S, Beyotime, Shanghai, China), and sperm T-AOC (Cat. BC1310, Solarbio, Beijing, China)) were identified in fresh, Sch B-treated, and non-treated sperm. Sperm MDA (lipid peroxidation marker) concentration was detected by a fluorescence enzyme labeler according to the manufacturer’s instructions, following the Tecan protocol. The absorbance was determined at 450 nm, 532 nm, and 600 nm. MDA measurements were presented in nmol/g protein. In addition, the activity of ROS enzymes and sperm T-AOC was measured in boar sperm, as previously analyzed. The experiment was repeated 3 times. ROS: relative fluorescence unit (RFU); MDA: nmol/mg; T-AOC: μmol/mg.

#### 2.4.7. Quantitative Real-Time PCR Analysis for SOD, CAT, GPx and Adhesion Protein Gene in Spermatozoa

The total RNA was extracted using the TRIzol reagent (Vazyme, Nanjing, China) for the boar sperm. The first-strand cDNA was synthesized from the total RNA using a reverse transcription kit (Cat. R211-02, Vazyme, Nanjing, China). Amplification reactions were conducted in triplicate using gene-specific primers designed from the clone sequences shown in Table 1. The PCR products in each cycle were monitored using a fluorescence quantitative PCR instrument (PRISMR 7500, ABI, Los Angeles, CA, USA). The data were analyzed using the comparison Ct (2^−ΔΔCt^) method and were expressed as fold changes relative to the respective controls. Each sample was analyzed in triplicate. The experiment was repeated 3 times.

#### 2.4.8. Measurement of Sperm Free Ca^2+^ Influx

Boar sperm was taken from each group after being cryopreserved and centrifuged with PBS 3 times (4 °C, 900 g/min, 5 min); then, the sperm density was adjusted to 2 × 10^6^/mL and centrifuged with 1/5 (4 °C, 900× *g*/min, 5 min). After the supernatant was abandoned, 300 μL PBS was added to the reinserted sperm. A 20 μL 0.1 mmol·L^−1^ Ca^2+^ fluorescent probe Fluo-3 (Cat. IF0150, Solarbio, Beijing, China) was added into the probe, which was fully blown with a pipette gun, and then incubated in a constant temperature incubator at 37 °C for 30 min. After centrifugation for 5 min (1500× *g*/min), the supernatant was abandoned and resuspended in 500 μL PBS. Then, the 200 μL sperm suspension was placed in a 96-well plate, and the fluorescence values of each experimental group were detected by a multifunctional enzyme marker [12]. The experiment was repeated 3 times. Ca^2+^: relative fluorescence unit (RFU).

#### 2.4.9. Lactic Acid Content in Spermatozoa

Lactic acid content was evaluated by using the lactic acid content assay kit (Cat. No. BC2230, Solarbio, Beijing, China). Samples were centrifuged at 3000× *g* for 5 min and the sperm pellets were re-suspended to a final concentration of 1 × 10^6^ cells in 0.5 mL of PBS. After ultrasonic fracturing in an ice bath, the samples were centrifuged at 8000× *g*/min for 10 min at 4 °C, and the supernatant was taken to measure the protein concentration and placed on ice until use. Then, 0.02 mL of distilled water, a standard tube, and the sample were added to the blank tube, standard tube, and determination tube, respectively, and then 1 mL enzyme working solution and 0.2 mL chromogenic agent were added, respectively, to mix evenly and react accurately at 37 °C for 10 min. Finally, 2 mL terminating solution was added. We mixed this evenly in a colorimetric dish with a 1 cm optical diameter and measured the absorbance of each tube at 530 nm [13]. The experiment was repeated 3 times. Lactic acid: nmol/g.

#### 2.4.10. Assessment of PKA and PKG in Boar Sperm by ELISA

Both PKA and PKG were identified in boar sperm using commercially available ELISA kits for PKA (Cat. MM-77570O1, Meimian, Jiangsu, China) and for PKG (Cat. MM-77540O1, Meimian, Jiangsu, China). We established the standard curve according to the kit’s instructions. Each test was performed in triplicate.

#### 2.4.11. Sperm Capacitation

As described in a previous study [14], after sperm collection, samples were centrifuged with PBS (4 °C, 900× *g*/min, 5 min) 3 times to wash away impurities, and then the sperm density was adjusted to 1 × 10^6^~2 × 10^6^ sperm/mL with Biggers–Whitten–Whittingham (BWW) capacitive medium, which was placed in a 5% CO_2_ cell incubator and incubated at 37 °C for 2 h.

#### 2.4.12. Data Analysis

Data were analyzed using one-way ANOVA (version 13.0, SPSS Inc., Chicago, IL, USA). When a significant difference (*p* < 0.05) was observed among treatments, Tukey’s studentized range test was used for post-test comparisons. Percent motility was arcsine-square root transformed, and the means of three straws per treatment were used for analysis. Values presented are mean ± SD.

## 3. Results

### 3.1. Experiment I

#### 3.1.1. Effects of Sch B on the Motility, Abnormal Sperm Morphology, Average Movement Velocity and Wobbility of Boar Sperm after Chilling

The effects of Sch B on motility, abnormal sperm morphology, average movement velocity, and wobbility at 4 °C were initially investigated in this sperm quality parameters analysis. Sperm motility is shown in Figure 1A. Sperm motility in 10 Sch B was significantly higher (*p* < 0.01), while 40 Sch B was markedly decreased (*p* < 0.001) compared to the 0 Sch B group. There was no significant difference between the 0 Sch B group and the Sch B-treated group (2.5, 5, and 20 Sch B) (*p* > 0.05).

The differences in abnormal sperm morphology are shown in Figure 1B. Compared to 0 Sch B, 5 Sch B and 10 Sch B significantly decreased the abnormal sperm morphology level in a dose-dependent manner. On the contrary, 40 Sch B significantly increased the abnormal sperm morphology level compared with 0 Sch B.

The effects of hypothermic injury on the average movement velocity of boar spermatozoa are shown in Figure 1C. There was no significant difference in the average movement velocity (*p* > 0.05) in 2.5 Sch B and 20 Sch B compared with 0 Sch B. The average movement velocity in 10 Sch B and 5 Sch B was significantly higher than 0 Sch B (*p* < 0.001 and *p* < 0.01, respectively) on the hypothermic liquid preservation. A significantly lower average movement velocity was observed in 40 Sch B than in 0 Sch B (*p* < 0.01)

The results of sperm wobbility after chilling are shown in Figure 1D. Our results showed that the addition of 10 μmol/L Sch B to the base solution can significantly improve the wobbility compared to 0 Sch B (*p* < 0.05), but there was no significant difference between the other Sch B-treated groups (2.5, 5, 20, and 40 Sch B) and 0 Sch B (*p* > 0.05).

#### 3.1.2. Effects of Sch B on the Acrosome Integrity and Plasma Membrane Integrity of Boars Sperm

After treatment with Coomassie bright blue, the acrosomal reaction rate of sperm was detected, as shown in Figure 2A. All Sch B-treated groups significantly (*p* < 0.01) enhanced plasma membrane integrity, except for 40 Sch B. When the Sch B concentration exceeded 10 μmol/L, the sperm acrosome integrity showed a decreasing trend with increasing concentration.

The percentage of plasma membrane integrity of sperm in the hypothermia treatment groups is presented in Figure 2B. The plasma membrane integrity in 2.5, 5, and 10 Sch B was significantly higher than in 0 Sch B (*p* < 0.01). It is worth noting that the plasma membrane integrity of boar sperm was significantly (*p* < 0.05) lower when the concentration of Sch B was 40 μmol/L.

#### 3.1.3. Effects of Sch B on the DNA Damage Rate and MMP Changes of Boars Sperm

The sperm DNA damage rate from representative semen samples of all six groups at the chilled–thawed stage is presented in Figure 3A. The DNA damage rate in 5, 10, and 20 Sch B was significantly lower than in 0 Sch B (*p* < 0.01). The difference between 0 Sch B and 2.5 Sch B was not significant (*p* > 0.05). When the concentration of Sch B was 40 μmol/L, the DNA damage rate showed an increasing trend.

Sperm MMP at the chilled-thawed stage are presented in Figure 3B. The percentage of sperm with MMP was higher (*p* < 0.01) in 10 Sch B than in 0 Sch B, while no significant difference was observed between 20 Sch B, 40 Sch B, and 0 Sch B after chilling (*p* < 0.05). The MMP was significantly lower in the 2.5 and 5 Sch B than in 0 Sch B (*p* < 0.01).

### 3.2. Experiment II

#### 3.2.1. Effects of Sch B on the *SOD*, *CAT* and *GPx* mRNAs Expression of Boar Sperm

The expression level of *SOD* and *CAT* was significantly increased in sperm with Sch B compared to 0 Sch B (*p* < 0.05, *p* < 0.01). The 0 Sch B showed a significant decrease (*p* < 0.001) in the mRNA level of sperm *SOD* and *CAT* compared to the fresh sperm (Figure 4). The differences in the quantitative expression of *GPx* mRNA were not significant in the fresh and Sch B-treated sperm compared to the non-treated sperm (*p* > 0.05).

#### 3.2.2. Effects of Sch B on the MDA, ROS and T-AOC Level of Boar Sperm

The 0 Sch B samples had significantly higher activity in sperm ROS compared to fresh sperm (*p* < 0.001; Figure 5A). The Sch B-treated samples revealed lower activity in sperm ROS (*p* < 0.01) compared to the 0 Sch B samples. When compared to fresh, 0 Sch B had a significantly lower T-AOC level (*p* < 0.001, Figure 5B). Compared to 0 Sch B, the Sch B-treated sperm had a significant increase (*p* < 0.001) in the T-AOC level. Compared to fresh, the 0 Sch B sperm had a significant increase in the sperm lipid peroxidation marker MDA (*p* < 0.001, Figure 5C), whereas the Sch B-treated sperm had a lower MDA content than the 0 Sch B sperm (*p* < 0.001).

#### 3.2.3. Effects of Sch B on The Ca^2+^ and Lactic Acid Content of Boars Sperm

Compared to fresh, 0 Sch B had significantly higher sperm Ca^2+^ and lactic acid contents (*p* < 0.01 and *p* < 0.001, respectively). Compared to 0 Sch B, Sch B-treated samples revealed significantly lower sperm Ca^2+^ and lactic acid contents (*p* < 0.05 and *p* < 0.01, respectively, Figure 6).

#### 3.2.4. Effects of Sch B on the mRNA Expression of *AWM, AQN1, AQN3, PSP-I* and *PSP-II* of Boar Sperm

The mRNA level of AWM was significantly reduced in 0 Sch B sperm compared to fresh (*p* < 0.001). However, boar sperm treated with Sch B at a dose of 10 μmol/L showed a significant increase (*p* < 0.01) in the mRNA level of AWM compared to 0 Sch B (Figure 7). The mRNA level of AQN1 was identified in fresh, Sch B-treated, and non-treated sperm. Both the non-treated and Sch B-treated sperm showed a significant reduction in the level of AQN1 mRNA compared to the fresh samples (*p* < 0.001) (Figure 7). No significant difference was observed between the non-treated and Sch B-treated samples (*p* ≥ 0.05). The mRNA levels of AQN3 in both the 0 Sch B and Sch B-treated sperm showed significant decreases compared to the fresh sperm (*p* < 0.01). AQN3 mRNA levels between the 0 Sch B and Sch B-treated sperm showed no significant differences (*p* ≥ 0.05). Quantitative expression of both PSP-I and PSP-II mRNA (Figure 7) showed significant increases in 0 Sch B compared to the fresh sperm (*p* < 0.001 and *p* < 0.01, respectively). When sperm were treated with Sch B at a dose of 10 μmol/L, a significant decrease in the quantitative expression of both PSP-I and PSP-II mRNA was identified compared to 0 Sch B (*p* < 0.01 and *p* < 0.05, respectively).

#### 3.2.5. Effects of Sch B on PKA and PKG Levels of Boar Sperm

PKA was increased in sperm without any additives compared to fresh (*p* < 0.05). However, boar sperm treated with 10 μmol/L Sch B showed a decrease in the PKA level compared to 0 Sch B as measured by ELISA immunoassay techniques (Figure 8A). Compared to the fresh, the 0 Sch B and 10 Sch B sperm both had no significant differences in PKG levels. PKG level between the 0 Sch B and Sch B-treated sperm showed no significant differences (*p* ≥ 0.05, Figure 8B).

### 3.3. Experiment III

#### 3.3.1. Effects of Sch B on Ca^2+^ and Lactic Acid Content in Capacitated Boar Sperm

Compared to 0 Sch B, the capacitation groups demonstrated significantly higher (*p* < 0.01) sperm Ca^2+^ and lactic acid contents. The Ca^2+^ and lactic acid contents in the 10 Sch B sample were not significantly different compared with the 0 Sch B capacitation sample (*p* > 0.05, Figure 9).

#### 3.3.2. Effect of Sch B on mRNA Expression of AWM, AQN1, AQN3, PSP-I, and PSP-II in Capacitated Boar Sperm

mRNA levels of both AWM and AQN1 were significantly reduced in the capacitation groups compared to 0 Sch B (*p* < 0.01). However, 10 Sch B showed no significant differences (*p* ≥ 0.05) in the AWM and AQN1 mRNA levels compared to 0 Sch B capacitation as measured by quantitative PCR (Figure 10). The mRNA levels of AQN3 in 0 Sch B capacitation and 10 Sch B showed no significant differences compared to 0 Sch B, as did 10 Sch B compared to 0 Sch B capacitation (*p* ≥ 0.05). Quantitative expression of both PSP-I and PSP-II mRNA (Figure 10) was significantly increased in 0 Sch B capacitation compared to 0 Sch B (*p* < 0.01 and *p* < 0.001, respectively). The 10 Sch B sample did not show significant differences with regard to the parameters of the two genes compared to the 0 Sch B capacitation sample.

#### 3.3.3. Effects of Sch B on PKA and PKG Levels in Capacitated Boar Sperm

The results of Sch B affecting sperm PKA and PKG levels of capacitated boar spermatozoa are shown in Figure 11. After 24 h of liquid storage, the sperm PKA levels of 0 Sch B capacitation and 10 Sch B were higher than 0 Sch B (*p* < 0.05), and the sperm PKA levels were not significantly different between 0 Sch B capacitation and 10 Sch B. Sperm PKG levels are shown in Figure 11. The PKG level in 0 Sch B was not significantly different compared with 0 Sch B capacitation (*p* > 0.05), and 0 Sch B capacitation showed no significant difference compared to 10 Sch B (*p* > 0.05).

## 4. Discussion

Sch B, isolated from Schisandra chinensis, has been documented to possess diversified pharmacokinetic properties, among them neurogenesis [15], anti-inflammatory [16], anti-tumor [17], and liver-protective [18] effects. The present study validated the effect of Sch B on male infertility [11], suggesting that Sch B is a potential natural active ingredient for reproduction protection. A previous study determined that oral administration of Sch B at 2 weeks not only increased the sperm number and average number of births but also increased sperm motility parameters in mice. These results suggest that Sch B may have positive effects on reproduction and improve sperm quality. Our present sperm quality parameters show that Sch B triggered an increase in the motility, average movement velocity, wobbility, plasma membrane integrity, and acrosome integrity of boar sperm, while decreasing abnormal sperm morphology, indicating that Sch B could directly act on sperm to enhance its motility. Notably, the plasma membrane integrity and acrosome integrity were increased after treatment with Sch B. Suggesting that Sch B has potential improvement functions on sperm quality parameters equally in in vitro models and at the cellular level, a previous study demonstrated that Sch B can modulate DNA damage through inhibition of MAPK/p53 signaling [19]. Our results are consistent with previous research on rodents in that administration with Sch B can decrease DNA damage of boar sperm in vitro. Subsequently, the results of the MMP changes showed that the MMP increase may be responsible for the decreased apoptosis and increased quality in Sch B-treated sperm. Our results correspond with a previous study in that the mitochondrial functional and structural integrity were improved by long-term Sch B treatment in rat kidneys [20]. These data corroborate the previous suggestion that Sch B increased the quality parameters by causing changes in anti-apoptotic ability, DNA damage reduction, and increased MMP. Sch B in concentrations of 0–10 μmol/L had a dose-dependent positive effect on sperm characteristics, while the higher levels of Sch B supplementation (20 and 40 μmol/L) had either a negative or no effect on sperm characteristics. Accordingly, we selected 10 μmol/L Sch B as the optimal concentration for a follow-up study.

Oxidative stress is a leading factor contributing to the poor overall quality of spermatozoa after the freezing and thawing process since the spermatozoa are rich in polyunsaturated fatty acids and are an easy target for ROS, which results in lipid peroxidation of the spermatozoan plasma membrane [21]. It has been well established in sperm that ROS content is inversely correlated with sperm motility [11]. Meanwhile, MDA is also involved in lipid peroxidation, which is an important indicator of cellular oxidative damage. The results of our analysis showed that the MDA and ROS contents were notably down-regulated and that the expression levels of *GPx* had no significance, accompanied by a significantly improved *SOD* and *CAT* mRNA expression and T-AOC levels after Sch B treatment in sperm, which is consistent with other studies performed in rodents, except for *GPx* [22,23]. Supplementation of extenders with *CAT*, *SOD*, and *GPx* can preserve sperm motility and DNA integrity, thereby improving cooled storage, especially with poor semen quality [24,25]. These results suggest that Sch B may improve sperm quality by increasing the antioxidant capacity of sperm.

Donà et al. reported that spermatozoon ROS content directly influences the levels and locations of tyrosine phosphorylation and then enables the spermatozoa to undergo an acrosome reaction [26]. In boars, the majority of seminal plasma proteins belong to the sperm-adhesin family, a group of (glyco) proteins built by a single CUB domain architecture [27]. Previous research has shown that the addition of seminal plasma attenuates in vitro and cooling-induced capacitation-like changes in boar spermatozoa [28]. Premature capacitation-like changes lead to spontaneous acrosome reactions and cell death, with a subsequent decrease in sperm fertilizing ability. The sperm adhesion protein gene family plays an important role in the processes of spermatogenesis, ejaculation, and sperm–egg binding. *AQN-1*, *AQN-3*, *AWN*, *PSP-I*, and *PSP-II* adhesion protein genes are the main members of the sperm adhesion protein gene family. The *AQN-1* and *AWN* proteins are involved in the formation of the acrosomal reaction inhibition receptor complex before sperm-egg binding and premature sperm capacitation can be prevented by them [29]. *AWN* and *AQN-3* also mediate sperm binding to the pellucida during fertilization [30]. It could be argued that the *PSP-I*/*PSP-II* heterodimer is responsible for the stabilization of the sperm membrane, thus counteracting the destabilizing phenomena that lead to membrane destabilization, impaired acrosome integrity, and reduced cell viability. Several studies have demonstrated a significantly higher expression of both genes in low-litter size boar spermatozoa than in high-litter size spermatozoa [31,32]. In addition, in another study, it was reported that the expression of PSP-I protein in seminal plasma was negatively correlated with the motility and morphology of boar sperm [33] and litter size [34]. Our results showed that Sch B could significantly increase the expression of *AWN*, whereas it reduced the expression of and decreased *PSP-I* and *PSP-II* in the boar sperm. However, the results of the expression of *AQN-1* and *AQN-3* were not significant. This is the first evidence revealing that the molecular mechanism of Sch B is involved in sperm capacitation by regulating the expression of the sperm adhesion protein gene family, suggesting that adding Sch B to a semen extender can improve sperm quality in the cryogenic liquid state by reducing capacitation.

Ca^2+^ signaling is of particular significance in sperm, where it is a central regulator in many key activities (including capacitation, hyperactivation, chemotaxis, and acrosome reaction). Previous studies have indicated that the increase in Ca^2+^ is related to sperm death. An excessive Ca^2+^ level affects capacitation during the fertilization of sperm, which leads to sperm death [35,36]. Fluorometric recordings of sperm loaded with fluo3-AM revealed that low-temperature preservation evoked elevations of intracellular Ca^2+^. Therefore, the high Ca^2+^ concentration during the cryopreservation of sperm is an important reason for its quality decline. In contrast, when boar spermatozoa were treated with Sch B in our study, decreases in the spermatozoa’s membrane instability and calcium concentration were observed. It has been reported that lactic acid is a sperm motility inactivation factor [37]. Carr concluded that the low pH of the cauda epididymal fluid from bulls and dogs, plus the presence of lactate, is sufficient to cause inhibition of sperm motility [38]. In our study, we found that the lactate content of sperm decreased after the addition of Sch B, which explained the possible factor of the previously detected increase in sperm motility, and we speculated that lactate content might be an important factor affecting sperm capacitation.

Sperm-capacitated processes shorten their lifespan and result in premature death [20]. Ejaculated spermatozoa are unable to fertilize the oocyte; on entry to the female reproductive tract, maturational changes occur in sperm that render them competent for fertilization in a process known as capacitation [39,40]. Even though the molecular basis of this process remains unclear, it correlates with a series of cellular and biochemical changes, including cholesterol efflux from the sperm plasma membrane [41], ion influx, membrane hyperpolarization [42,43], cAMP production [44], kinase activation and protein phosphorylation [45]. Enzymes such as members of the PKA are directly regulated by cAMP [44]. Thus, this nucleotide orchestrates several downstream events with critical outcomes in cell physiology. Activation of PKA enzymes is essential for capacitation, and thus, cAMP levels are tightly regulated during this process. cAMP, an important second messenger, activates PKA, leading to potassium channel opening and calcium channel closing, which can decrease the concentration of calcium in the cytoplasm. Previous results suggest that [46] Sch B was able to significantly improve erectile dysfunction in rats with nerve injury through the cAMP-PKA pathways. Similar results were found in our present capacitation investigation in that Sch B triggered an increase in the PKA level, resulting in a Ca^2+^ concentration decrease. PKG inhibitors have been reported to inhibit sperm, hyperactivation, and the acrosome reaction. It has also been reported that increased PKG in the cGMP/PKG signaling pathway can increase Ca^2+^ and tyrosine-phosphorylated proteins, promote hyperactivation, and induce the acrosome reaction, which ultimately facilitate sperm capacitation [47]. Our study showed that adding Sch B had no effect on the level of PKG in boar sperm. We believe that adding Sch B in a semen extender can regulate sperm Ca^2+^ concentration through the cAMP-PKA pathway, which leads to the decrease of sperm number with capacitation. These findings suggest that Sch B was identified as a primary candidate for a decapacitation factor.

Since our data showed that Sch B could affect the capacitation state of sperm by regulating sperm adhesion proteins and the CAMP-PKA pathway, we therefore hypothesized that Sch B may be involved in decapacitation. To test this hypothesis, we further assessed Ca^2+^ influx, lactic acid content, the quantitative expression of sperm adhesion protein gene family mRNAs, and the PKA/PKG activity of boar spermatozoa after capacitation. However, no significant difference (*p* > 0.05) was observed in any of the capacitation-related parameters (Ca^2+^ and lactic acid content, expression of sperm adhesion protein gene family, PKA and PKG levels) evaluated in samples with Sch B compared to 0 Sch B. Current studies on sperm decapacitation factors have shown that they can prevent premature capacitation and the acrosome reaction but have no decapacitation effect on sperm that has been capacitated [48,49]. In other words, Sch B does not reverse capacitation when spermatozoa are highly capacitated. Therefore, we suspected that Sch B acted only on non-capacitated sperm, as it did not reverse capacitated sperm. This selectivity allows its decapacitating action only in non-capacitated sperm, without affecting capacitated sperm.

Collectively, our study based upon sperm quality parameters, antioxidant capacity, and sperm fertilization ability during liquid preservation of boar semen at 4 °C significantly supports the effect of Sch B at a dose of 10 μmol/L as a semen storage protectant. This was evidenced by downregulation of Ca^2+^, reduced PKA, and regulated sperm adhesin family, reduced amounts of capacitated sperm, elevated antioxidative levels (SOD, CAT, and T-AOC increased, ROS, and MDA decreased), lowered proapoptotic properties of DNA damage, increased MMP, and, ultimately, the protected quality of boar sperm. Based on the above results, Sch B was identified as a primary candidate for an antioxidant and decapacitation factor in sperm in liquid at 4 °C.

## 5. Conclusions

In conclusion, our findings demonstrated that 10 μmol/L is the optimal concentration of Sch B supplement in boar semen extender. Sch B enhanced sperm quality by increasing oxidation resistance and reducing the capacitation-like changes in spermatozoa stored at 4 °C. However, it did not have reversing effects on the capacitation status of capacitated spermatozoa.

## Figures and Tables

**Figure 1 animals-13-00848-f001:**
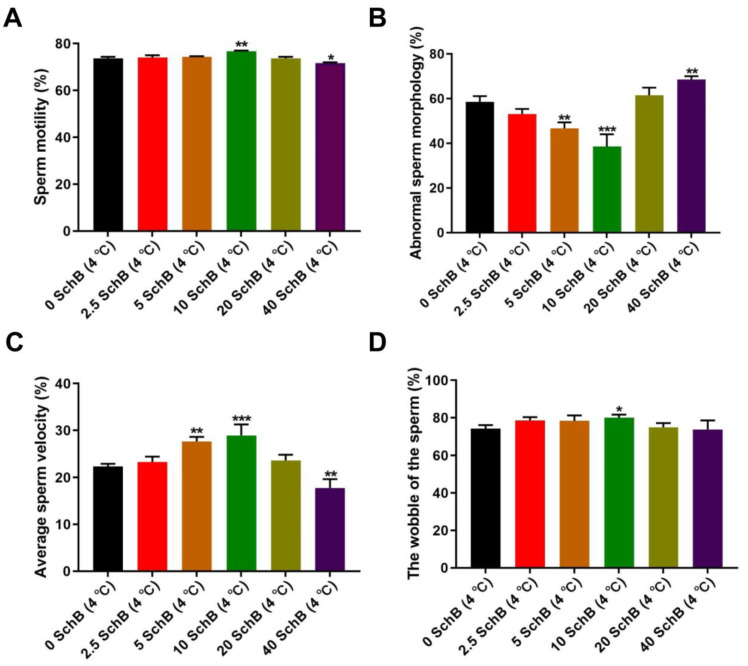
(**A**–**D**): Effect of Sch B on the motility, abnormal sperm morphology, average movement velocity and wobbility of boar sperm: (**A**) Sperm motility. (**B**) abnormal sperm morphology (**C**) Average movement velocity. (**D**) Sperm wobbility. Data are expressed as mean ± SD (*n* = 12). *** *p* < 0.001, ** *p* < 0.01 and * *p* < 0.05 vs. 0 Sch B.

**Figure 2 animals-13-00848-f002:**
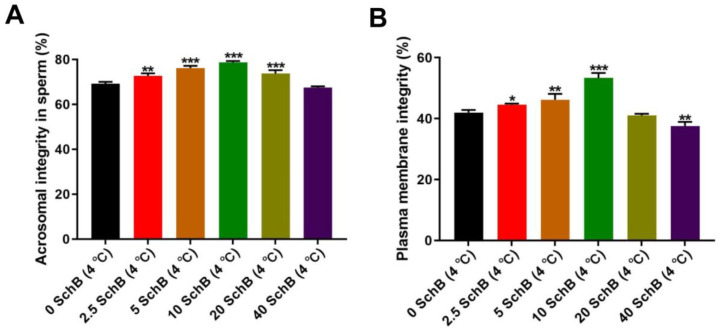
(**A**,**B**): Effects of SchB on the plasma membrane integrity and acrosome integrity of boar sperm: (**A**) Sperm motility. (**B**) abnormal sperm morphology. Data are expressed as mean ± SD (*n* = 12). *** *p* < 0.001, ** *p* < 0.01 and * *p* < 0.05 vs. 0 Sch B.

**Figure 3 animals-13-00848-f003:**
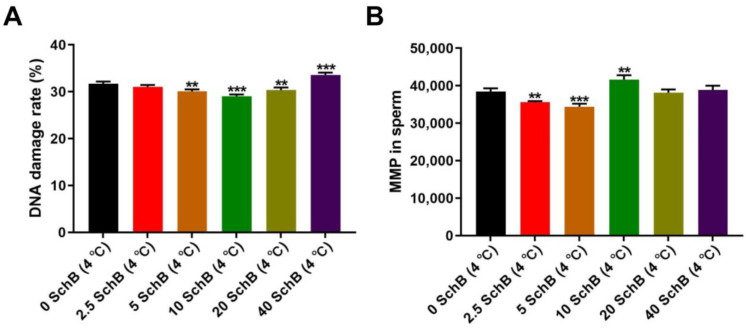
Effects of Sch B on the DNA damage rate and MMP changes in boar sperm: (**A**) DNA damage rate. (**B**) MMP changes in sperm. Data are expressed as mean ± SD (*n* = 12). *** *p* < 0.001 and ** *p* < 0.01 vs. 0 Sch B.

**Figure 4 animals-13-00848-f004:**
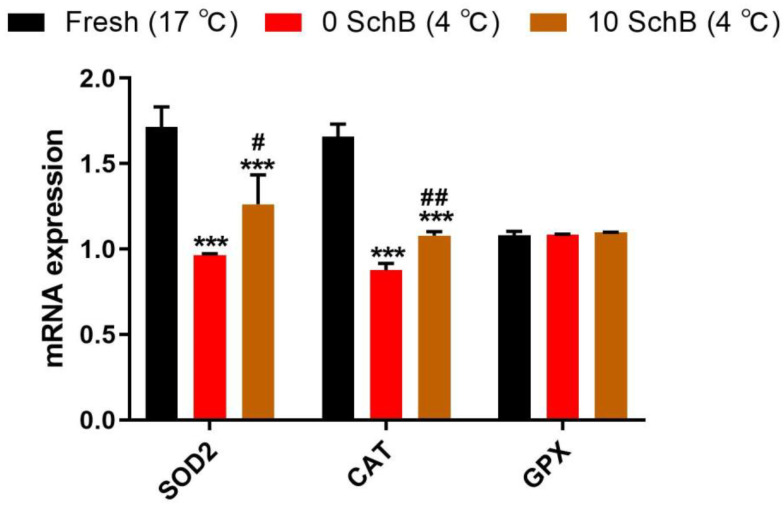
Effect of Sch B on mRNA and of *SOD, CAT,* and *GPx* in boar sperm: Data are expressed as mean ± SD (*n* = 12). ## *p* < 0.01 and # *p* < 0.05 vs. 0 Sch B, *** *p* < 0.001 vs. fresh.

**Figure 5 animals-13-00848-f005:**
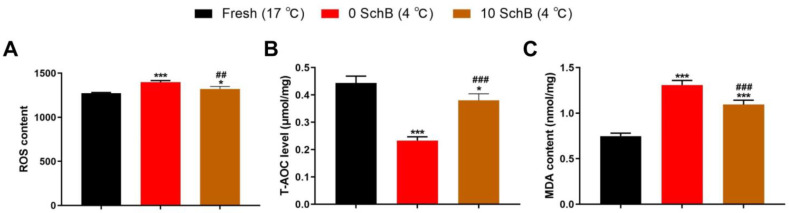
Effect of Sch B on boar sperm ROS (**A**), T-AOC (**B**), and MDA level (**C**): Data are expressed as mean ± SD (*n* = 12). ### *p* < 0.001 and ## *p* < 0.01 vs. 0 Sch B, *** *p* < 0.001 and * *p* < 0.05 vs. fresh.

**Figure 6 animals-13-00848-f006:**
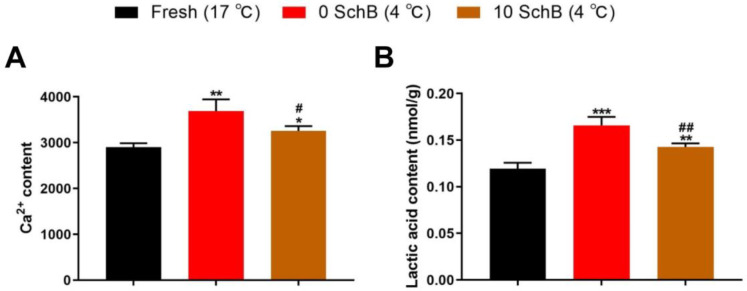
Effects of Sch B on boar sperm Ca^2+^ (**A**) and lactic acid (**B**) content: Data are expressed as mean ± SD (*n* = 12). ## *p* < 0.01 and # *p* < 0.05 vs. 0 Sch B, *** *p* < 0.001 ** *p* < 0.01 and * *p* < 0.05 vs. fresh.

**Figure 7 animals-13-00848-f007:**
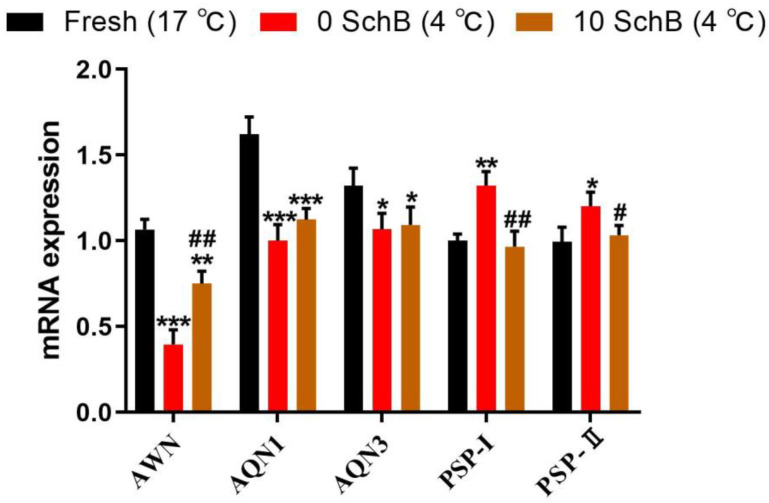
Effects of Sch B on boar sperm mRNA expression of *AWM, AQN1, AQN3, PSP-I,* and *PSP-II* in boar sperm: Data are expressed as mean ± SD (*n* = 12). ## *p* < 0.01 and # *p* < 0.05 vs. 0 Sch B, *** *p* < 0.001 ** *p*< 0.01 and * *p* < 0.05 vs. fresh.

**Figure 8 animals-13-00848-f008:**
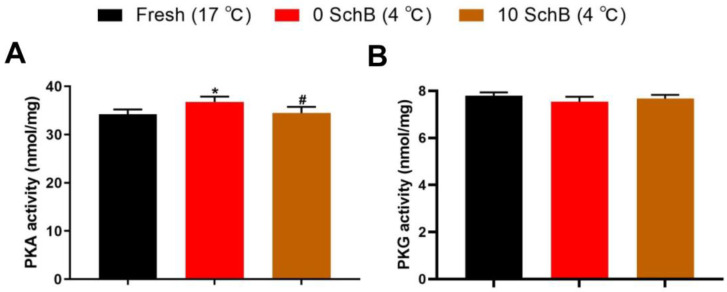
Effects of Sch B on PKA (**A**) and PKG Levels (**B**) in boar sperm: Data are expressed as mean ± SD (*n* = 12). # *p* < 0.05 vs. 0 Sch B, * *p* < 0.05 vs. fresh.

**Figure 9 animals-13-00848-f009:**
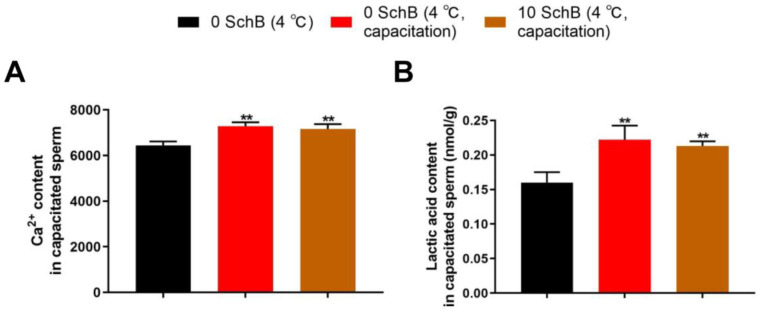
Effects of Sch B on Ca^2+^ (**A**) and lactic acid (**B**) content in capacitated boar sperm: Data are expressed as mean ± SD (*n* = 12). ** *p* < 0.01 vs. 0 Sch B.

**Figure 10 animals-13-00848-f010:**
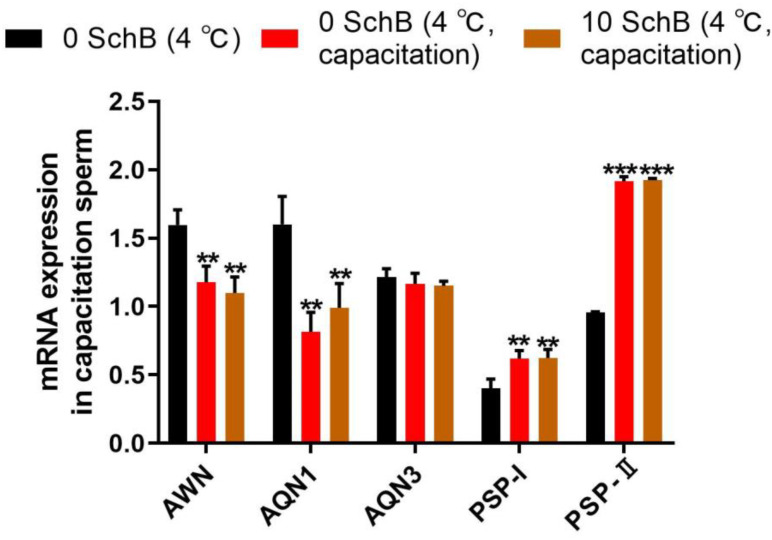
Effects of Sch B on mRNA expression of AWM, AQN1, AQN3, PSP-I, and PSP-II in capacitated boar sperm: Data are expressed as mean ± SD (*n* = 12). *** *p* < 0.001 and ** *p* < 0.01 vs. 0 Sch B.

**Figure 11 animals-13-00848-f011:**
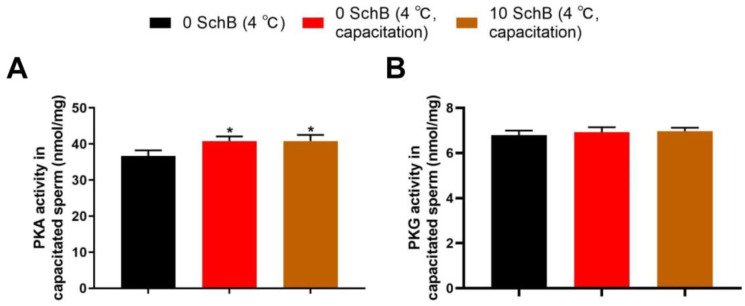
Effects of Sch B on PKA (**A**) and PKG Levels (**B**) in capacitated boar sperm: Data are expressed as mean ± SD (*n* = 12). * *p* < 0.05 vs. 0 Sch B.

**Table 1 animals-13-00848-t001:** List of primer sequences used in real-time PCR analysis.

Gene	Primer Sequences
*SOD2*	F:GCTGGAAGCCATCAAACG
	R:TTAGAACAAGCGGCAATCTG
*CAT*	F:TGCCCATACTTCCCGTCC
	R:GGTCCAGGTTACCGTCAG
*GPx*	F:CAGGTACAGCCGTCGCTTTC
	R:AAAATCCCGAGAGTAGCACT
*AQN1*	F:GCTGCTCAGTACAGCTACAC
	R:ATTACCAGCACCTGGCAGTC
*AWN*	F:CAATACCGCCTCTCAACCTC
	R:GGCCTTCTGGATCAGCATAG
*AQN3*	F:AACGGGCAGGACCTGATAAC
	R:CCCTCAGACACGAGTCTAAG
*GAPDH*	F:CCCCAACGTGTCGGTTGT
	R:CTCGGACGCCTGCTTCAC

## Data Availability

The data presented in this study are available on request from the corresponding author.

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
