# Peer review of "Supplementation of Schisandrin B in Semen Extender Improves Quality and Oxidation Resistance of Boar Spermatozoa Stored at 4 °C"

_animals, 2023, doi:10.3390/ani13050848_

Round 1
Reviewer 1 Report
The authors carry out an interesting and very complete study on the use of Schisandrin B at various concentrations in the conservation of boar semen at 4ºC, demonstrating that the use of 10 µmol/L significantly improves a large number of parameters of viability and sperm quality analyzed and also in capacitated semen.
It is well known that boar semen is routinely kept diluted in refrigeration at 15ºC for 3-7 days (depending on the diluent used) and that frozen boar semen in the freezing process has serious viability and post-thawing quality problems, so that it can be used ritually in A.I. Normally, in the laboratory, the temperature of 4ºC is used as the transition temperature, from 25º or 15ºC, to start the freezing process, and in most protocols, the semen remains at 4º-5ºC for 2 to 4 hours at this temperature, before starting the temperature drop to freeze the samples manually or automatically. It would be of great interest to review whether the use of Schisandrin B also protects porcine sperm during the freezing-thawing process.
I think that it should be accepted for publication, but, nevertheless, a series of major and minor considerations should be made to improve the publication. Some are of greater significance and others of lesser importance, they are considerations for revising the text to avoid possible errors and improve its edition.
Major considerations:
1. Title: The title of the paper should indicate that the use of Schisandrin B has promoted a beneficial effect on boar semen stored at a refrigerated temperature of 4ºC, since its effect has not been proven at a temperature of 15ºC, nor in frozen semen.
2. "Simple Summary" is missing. 3. In “Material and Methods” section, many details of interest are missing: - A section on "Experimental Design" is very necessary at the beginning of the “Material and Methods”.- It does not detail the age of the males or their feeding.
- How often are semen samples collected from boars?
- The samples used in the test were from each male or a mixture of several was prepared to eliminate the individual effect?
- It is not explained how the seminal samples were collected until they were used in the laboratory, nor how they were diluted nor the diluent used.
- The storage time of the seminal samples at 4ºC is not detailed and this is essential.
- The number of replicates in many of the different experimental tests (2.1, 2.2, 2.3, 2.5, 2.6, 2.8 and 2.9) are not explained.
- In “Results”, the different tests are grouped into three experiments that do not have their proper correspondence in “Material and Methods” due to the lack of an adequate experimental design.
- It is not explained in detail how the spermatozoa have been capacitated experimentally, nor is it previously mentioned that these tests will be carried out.
- I once again emphasize the need for an adequate and explanatory “Experimental Design” section to facilitate the reader's understanding of the paper.
Minor considerations:
1. The term “wobbility” must be explained (page 1, line 22; page 2, line 71).
2. “Abbreviations” must be explained in the text as well as in Abstract, such as “Sch B” (page 2, line 53), and some of them are not explained ("BWW” page 5, lines 157,158). Also MMP and T-AOC.
3. "Introduction" section is very short and concise.
4. The number of animals and the breed only appears in “Abstract” and must be included in “Material and Methods”.
5. All references to centrifugations must be in the same unit, preferably in “g”, so replace those that use only "r", for example (page 2, lines 73, 87).
6. In “Results” section, the units of measurement of many of the parameters analyzed are missing. Thus, these measurement units are not detailed in Figures 2A, 2B, 3B, 4, 5A, 7, 8, 9, 10 and 11. In figures 5B, 5C, 6B and 9B "prot" must be eliminated or its meaning explained at the bottom of the figure.
7. “the motility” (page 6, line 193).
8. “Gpx” or “GPx” (page 7, lines 230, 234; page 12, line 362).
9. “p” or “P” (page 7, line 235).
10. “boar sperm” (page 7, line 237).
11. “Carr et al.” (page 13, line 405).
Reviewer 2 Report
I want to thank the author for the work done. The study is well designed and well described.
I have some comments that I would like to be considered.
First, the English is very bad and needs extensive editing. Otherwise, the ideas are there well described and the flow of the discussion is well-written,
1. Provide more description of Sch B is it a plant, or extracted from petroleum...etc, country of origin where it was bought how it was extracted and prepared for the study...etc
2. line 63 what do you mean by inoxidizability please change the word
3. Material and methods:
Give more details about the animals how many? how did you prepare the semen which extender used in this study ...etc.
Where did you buy the SChB ...etc
How many frames did you analyse the motility ?
What did you mean by motility; total motility, progressivity...etc
What do you mean deformity rate ? sperm morphoabnormality.?..
Please put reference for the different methodologies used when needed like sperm free CA influx and Lactic acid content....etc
Line 207 reformulate the sentence, please.
4.Results:
Some figures are bigger than others please try to use the same size for all
Discussion:
Compare the effect of SCh B on sperm when given as a treatment(orally) may be influenced by many factors. Better to not use a direct comparison.
5.Conclusion: rewrite the conclusion, please. you have very interesting results that you can give them a better value.
Round 2
Reviewer 1 Report
Line 47: I consider that 4ºC cannot be considered as cryopreservation, that temperature is refrigeration. That sentence should be corrected as it can be misleading.
Author Response
We followed the comment and replaced the cryopreservation in line 47 with chill storage.